# A multi-gene predictive model for the radiation sensitivity of nasopharyngeal carcinoma based on machine learning

Kailai Li[1†], Junyi Liang[1†], Nan Li[2†], Jianbo Fang[1†], Xinyi Zhou[1], Jian Zhang[1*], Anqi Lin[1*], Peng Luo[1*], Hui Meng[1*]

[1]Department of Oncology, Zhujiang Hospital, Southern Medical University, Guangzhou, China; [2]Department of Radiation Oncology, Nanfang Hospital, Southern Medical University, Guangzhou, China

*For correspondence:
zhangjian@i.smu.edu.cn (JZ);
smulinanqi0206@i.smu.edu.cn (AL);
luopeng@smu.edu.cn (PL);
menghui8816@163.com (HM)

[†]These authors contributed equally to this work

Competing interest: The authors declare that no competing interests exist.

## eLife Assessment

The authors have developed a robust machine learning approach to predict radio sensitivity in patients with NPC based on a defined gene signature. Some key aspects of this signature have been validated in vitro using relevant cell lines which strengthens the conclusions of this **important** and **convincing** study. The publication will be of interest to clinicians working on this indication as well as a more broader readership made up of scientists working on radiation biology and those with a bioinformatics/machine learning background.

**Abstract** Radiotherapy resistance in nasopharyngeal carcinoma (NPC) is a major cause of recurrence and metastasis. Identifying radiotherapy-related biomarkers is crucial for improving patient survival outcomes. This study developed the nasopharyngeal carcinoma radiotherapy sensitivity score (NPC-RSS) to predict radiotherapy response. By evaluating 113 machine learning algorithm combinations, the glmBoost+NaiveBayes model was selected to construct the NPC-RSS based on 18 key genes, which demonstrated good predictive performance in both public and in-house datasets. The study found that NPC-RSS is closely associated with immune features, including chemokine factors and their receptor families and the major histocompatibility complex (MHC). Gene functional analysis revealed that NPC-RSS influences key signaling pathways such as Wnt/β-catenin, JAK-STAT, NF-κB, and T cell receptors. Cell line validation confirmed that SMARCA2 and CD9 gene expression is consistent with NPC-RSS. Single-cell analysis revealed that the radiotherapy-sensitive group exhibited richer immune infiltration and activation states. NPC-RSS can serve as a predictive tool for radiotherapy sensitivity in NPC, offering new insights for precise screening of patients who may benefit from radiotherapy.

## Introduction

Nasopharyngeal carcinoma (NPC), which originates from the nasopharyngeal mucosal epithelium, is a malignant tumor with distinct epidemiological, histopathological, clinical, and therapeutic features among head and neck tumors (*Chua et al., 2016*; *Ferlay et al., 2015*; *Wei and Sham, 2005*). According to the most recent global cancer statistics, a total of 133,354 new cases of nasopharyngeal cancer were reported, accounting for only 0.7% of all new cancer cases that year (*Sung et al., 2021*). Although the incidence of nasopharyngeal cancer is relatively low compared with other cancer types, it has a unique geographic distribution. About 70% of new nasopharyngeal cancer cases occur in Eastern and Southeastern Asia, while the rest are mainly in South-Central Asia, North Africa, and East Africa

(*Chen et al., 2019*). Currently, radiotherapy is used as the primary curative treatment for patients with NPC. With the continuous advancement of radiotherapy technology, recent studies have shown that hyperfractionated intensity-modulated radiotherapy increases the 3-year overall survival rate in these patients from 55.0% to 74.6% compared to conventional fractionated intensity-modulated radiotherapy (*You et al., 2023*). However, approximately 20–30% of patients experience locoregional recurrence of the tumor due to radiation resistance (*Lee et al., 2015*; *Ma and Chan, 2005*; *Agulnik and Epstein, 2008*). Studies have shown that radiotherapy resistance is the main reason for treatment failure in nasopharyngeal cancer patients (*Linkous and Yazlovitskaya, 2012*; *Ribassin-Majed et al., 2017*). Given the importance of radiotherapy in nasopharyngeal cancer treatment, it is imperative to investigate the molecular mechanisms of radiotherapy resistance and develop strategies to enhance the radiosensitivity of nasopharyngeal cancer cells.

In recent years, artificial intelligence (AI) has been deeply integrated into the medical field (*Khan, 2023*). Machine learning (ML), an important branch of AI, has the ability to process nonlinear data, discover new patterns, and generate predictive models after learning from existing data. Machine learning plays a crucial role in developing predictive markers for tumor treatment and prognosis. Recent studies have focused on utilizing machine learning techniques to construct predictive models for these purposes. These studies employ various methods and algorithms, including integrated learning, immune-derived feature extraction, and AI network-guided signatures. For instance, one study utilized machine learning through tumor MR image feature mapping for patient stratification in adjuvant chemotherapy for locally advanced NPC (*Teng et al., 2023*). Another study employed machine learning based on MRI/DWI radiomics signatures to predict the prognosis of NPC (*Hu et al., 2022*). Additionally, machine learning-based algorithms have been developed to create predictive models for identifying lncRNAs associated with tumor-infiltrating immune cells, aiming to improve prognosis and immunotherapy response in cancer patients (*Liu et al., 2022*; *Zhang et al., 2022*). These studies aim to enhance patient outcomes and prognosis while providing superior predictive power for personalized clinical treatment strategies. However, there is still a lack of machine learning-based biomarkers for predicting radiotherapy sensitivity in NPC.

In this study, a combination of multiple machine learning methods and transcriptomic data from nasopharyngeal cancer patients was used to construct a biomarker that can predict the sensitivity to radiotherapy in nasopharyngeal cancer patients, termed the Nasopharyngeal Carcinoma Radiosensitivity Score (NPC-RSS). In addition, we explored the biological mechanisms underlying the relationship between NPC-RSS and radiotherapy response in nasopharyngeal cancer patients. We anticipate that the construction of NPC-RSS will guide the selection of radiotherapy strategies for NPC patients, thereby improving the clinical outcomes and prognosis of these patients.

## Results

### Construction of the NPC-RSS model

As shown in *Figure 1*, 48 models were successfully constructed, and the area under the curve (AUC) of each model was calculated in both the training and validation sets. The best model combination was found to be glmBoost + NaiveBayes after sample weighting: the glmBoost algorithm was used to screen out the most valuable gene features, and then the NaiveBayes algorithm was employed to identify the most reliable model (*Figure 2A*). The model identified 18 genes, which were used as candidate genes for subsequent studies. The NPC-RSS was constructed using logistic regression, and the model prediction formula after secondary modeling was: P(Y=1)=−15.3906 + 2.7369 * C1QTNF3+3.8292 * CA11 + 7.6856 * CD9 - 15.9445 * CDK5RAP3 - 11.2475 * CLDN1 + 14.3984 * DMC1 - 6.2474 * EYA1 + 1.6897 * IFI44L+10.0473 * KNG1 + 0.9464 * KREMEN1 - 6.5371 * NT5DC2 - 10.4192 * NTRK3 + 8.1306 * PSG4 - 0.5755 * RFX4 - 2.1728 * RHOBTB3 - 7.1597 * SLC1A2+22.6640 * SMARCA2 - 0.3858 * TRIM58. The top 5 genes characterized in the NPC-RSS, in order of their contribution to radiotherapy sensitivity and weighting coefficients, are SMARCA2, DMC1, CD9, PSG4, and KNG1. The genes with absolute values of weight coefficients greater than 10 were SMARCA2, DMC1, KNG1, NTRK3, CLDN1, and CDK5RAP3 (*Figure 2B*).

The prediction model constructed using these 18 genes demonstrated good diagnostic efficacy, with an area under the receiver operating characteristic (ROC) curve (AUC) of 0.996 for the training set. The GSE32389 dataset was employed as an external validation set, yielding an AUC of 0.823,

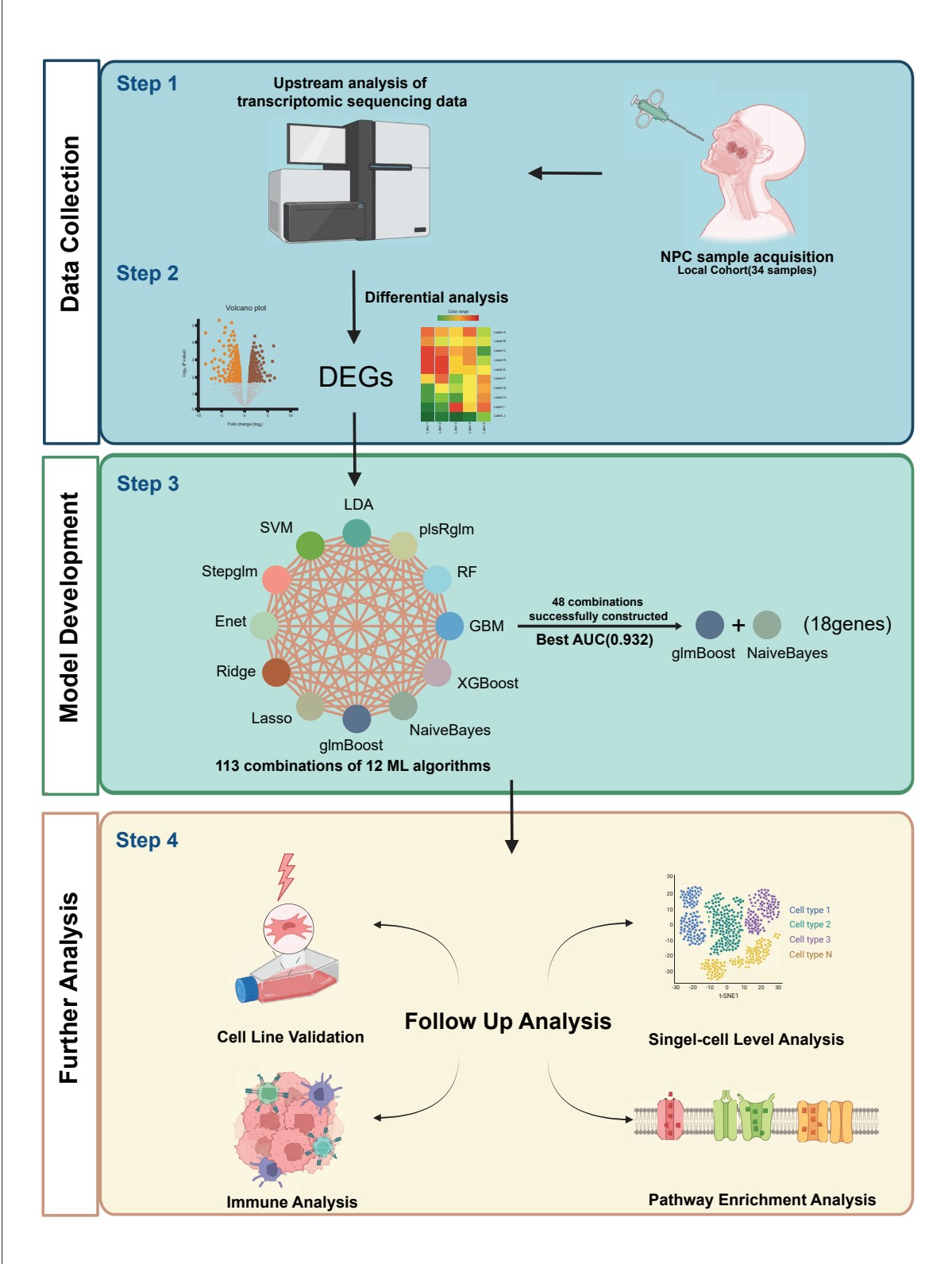

**Figure 1.** Flowchart for constructing a predictive model for NPC radiotherapy sensitivity. Differentially expressed genes obtained from local NPC transcriptome data, grouped according to radiosensitivity and radioresistance, were used to predict radiotherapy sensitivity scores (NPC-RSS) of NPC patients using 12 machine learning algorithms, including Lasso, Ridge, Enet, Stepglm, SVM, glmBoost, LDA, plsRglm, RandomForest, GBM, XGBoost, and NaiveBayes. Additionally, 48 other combinations of validated frameworks were constructed to predict the radiotherapy sensitivity score (NPC-RSS)

*Figure 1 continued on next page*

*Figure 1 continued*

of nasopharyngeal carcinoma patients. The most effective NPC-RSS was finally constructed based on the combination of glmBoost + NaiveBayes, which yielded the best AUC. The role and biological significance of NPC-RSS in NPC radiotherapy sensitivity were comprehensively explored through tumor immune microenvironment analysis, pathway enrichment analysis, and single-cell transcriptomic analysis.

The online version of this article includes the following source data for figure 1:

**Source data 1.** Gene expression data from RNA-seq.

which demonstrated the model's robustness. The overall sample-weighted AUC for the combined training and validation sets was 0.932 (*Figure 2C*). The genes used to construct the model were derived from our center's NPC transcriptome expression data and underwent differential expression analysis, resulting in 2932 differentially expressed genes (DEGs; *Figure 2D*).

Our previous analysis revealed that 18 gene features were consistently expressed in the NPC dataset. Furthermore, these 18 gene features exhibited significant differential expression between the sensitive and resistant groups (*Figure 2—figure supplement 1*).

### In vitro validation of NPC-RSS

The induction process of CNE2-RS radiation-resistant cells is shown in *Figure 2E*. RNA was transcribed into cDNA, sequenced, and analyzed upstream by a high-throughput sequencer to obtain count data, which was further transformed into FPKM values for subsequent analysis. Differential analysis was performed between the parental and resistant groups using the limma package. The analysis results showed that SMARCA2 and CD9, among the key genes of NPC-RSS, were significantly highly expressed in the sensitive group (p<0.001, t-test), which was consistent with the expression trend of the key genes in the previously constructed NPC-RSS (*Figure 2F*). The heatmap of the differential analysis is shown in *Figure 2G*. We also clustered these data based on the 18 genes in NPC-RSS after Z-score normalization for the heatmap. The results were consistent with the expression trend in NPC-RSS (*Figure 2H*). Meanwhile, we calculated the radiotherapy sensitivity scores of three pairs of replicated CNE2-P and CNE2-RS samples using NPC-RSS and used the mean values for statistical testing. The results showed that the sensitivity scores of the sensitive group were significantly higher than those of the resistant group, further confirming the reliability of NPC-RSS (*Figure 2I*).

### Annotation of key NPC-RSS genes and immune-related features

To thoroughly explore the potential mechanism by which the NPC-RSS predicts radiosensitivity, we stratified the local NPC data into groups based on the NPC-RSS developed in this study and examined the association between these groups and immune infiltration. The proportion of immune cell content in each patient and the correlations among immune cells were visualized using various formats. The results demonstrated a significant elevation in the levels of T follicular helper cells, regulatory T cells (Tregs), and activated NK cells in the samples from the radiosensitive group compared to those from the radioresistant group (*Figure 3A*), suggesting that the radiosensitive group exhibited a more immunologically active infiltration profile.

In this study, we further explored the relationship between NPC-RSS key genes and immune cells. We found that several key genes were highly correlated with immune cells: SMARCA2 was significantly positively correlated with resting CD4 + memory T cells; DMC1 was significantly positively correlated with resting NK cells; CD9 was significantly positively correlated with resting CD4 + memory T cells; and KNG1 was significantly negatively correlated with M1 macrophages, γδ T cells, monocytes, eosinophils, neutrophils, and other immune cell types (*Figure 3B*). Furthermore, all these immune cell types were significantly correlated with each other (*Figure 3C*). Correlations between these five key genes and various immune factors, including immunomodulators, chemokines, and cellular receptors, were obtained from the TISIDB database (*Figure 3D*). These analyses suggest that the NPC-RSS key genes are closely associated with the level of immune cell infiltration and play a crucial role in the immune microenvironment.

### Validation of NPC-RSS at the single-cell level

The expression patterns of key NPC-RSS genes at the single-cell level exhibited a stronger correlation with model predictions. We analyzed the single-cell transcriptome profiles of four patients. Among

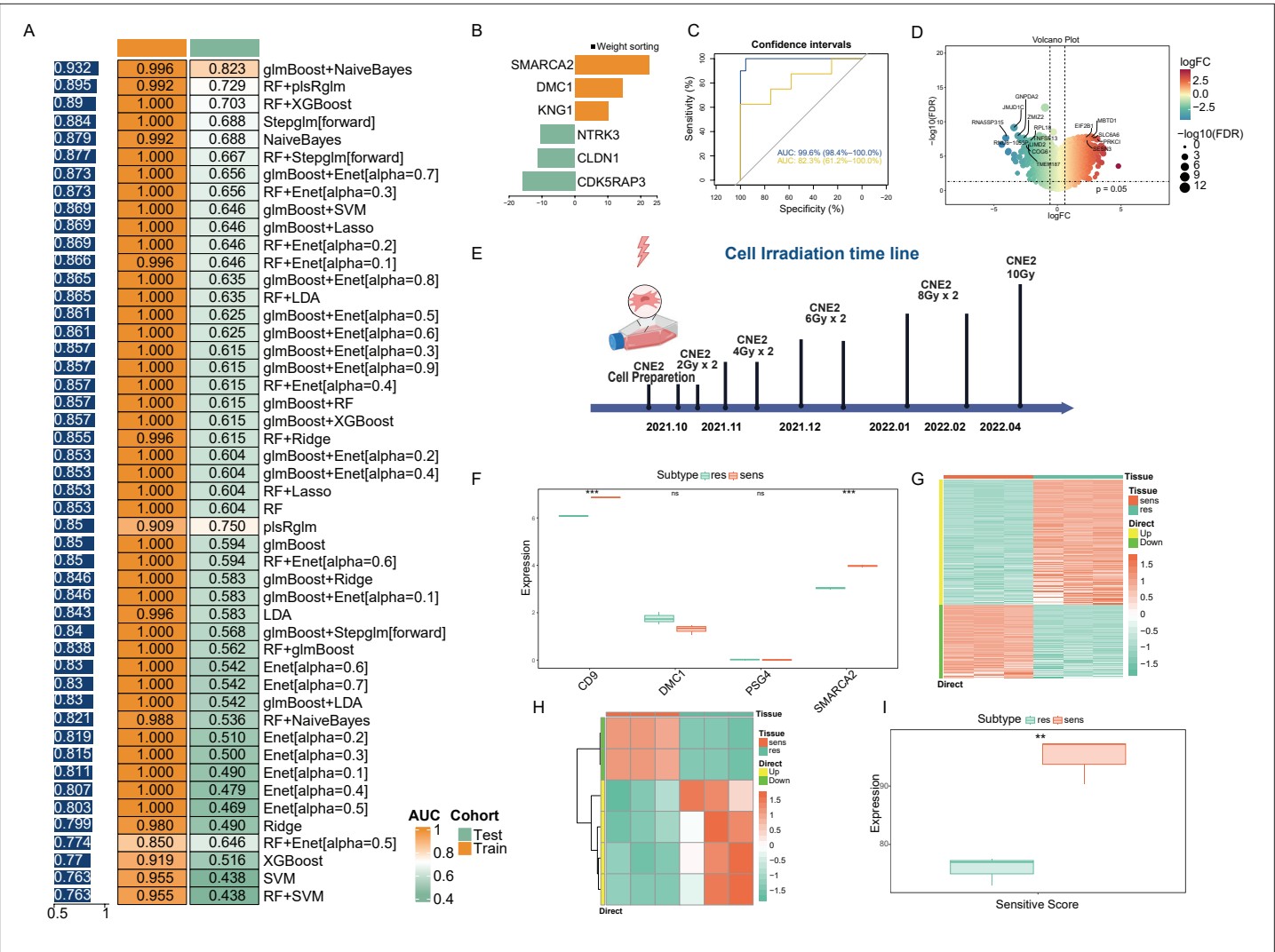

**Figure 2.** Consensus NPC-RSS construction and validation using an integrated machine learning approach. (**A**) Based on 48 combined validation frameworks, including 12 machine learning algorithms (Lasso, Ridge, Enet, Stepglm, SVM, glmBoost, LDA, plsRglm, RandomForest, GBM, XGBoost, and NaiveBayes), the area under the curve (AUC) of each model was calculated for both the training (n=34) and validation sets (n=20). Heatmaps were generated based on the sample-weighted AUC values. (**B**) Demonstration of NPC-RSS genes with absolute values of weight coefficients greater than 10, based on the glmBoost and NaiveBayes combination. (**C**) Receiver operating characteristic (ROC) curves for the training and validation sets. (**D**) Volcano plot depicting differentially expressed genes between the sensitive and resistant groups of the CNE-2 cell line. (**E**) Timeline of resistance strain induction in the CNE-2 cell line. Expression of the top 5 weighted genes in the NPC-RSS signature for the CNE-2 cell line. Data are presented as mean ± SD (n=3 biological replicates). Statistical significance was determined by Student's t-test (*p < 0.05, **p < 0.01, ***p<0.001). (**G**) Heatmap of differentially expressed genes between the sensitive and resistant groups of the CNE-2 cell line. (**H**) Heatmap of z-scores for NPC-RSS genes in the CNE-2 cell line. (**I**) Analysis of differences in sensitivity scores among CNE-2 cell lines. Data are presented as mean ± SD (n=3 biological replicates). *p < 0.05, **p < 0.01, ***p<0.001.

The online version of this article includes the following figure supplement(s) for figure 2:

**Figure supplement 1.** Expression of 18 model genes grouped according to NPC-RSS in our center's transcriptome data.

the four patients, three were sensitive to radiotherapy, while one was resistant. Following single-cell suspension preparation, library construction, sequencing analysis, and quality control of the obtained data, we acquired transcriptome data for a total of 28,957 cells. Among these, 22,796 cells were derived from the radiotherapy-sensitive group, and 6161 cells were from the radiotherapy-resistant group. Unsupervised clustering was performed on the 28,957 cells obtained from sequencing to identify cell subpopulations based on the similarity of their gene expression profiles. As illustrated in *Figure 4*, 17 cell subpopulations were identified through clustering. By examining the expression

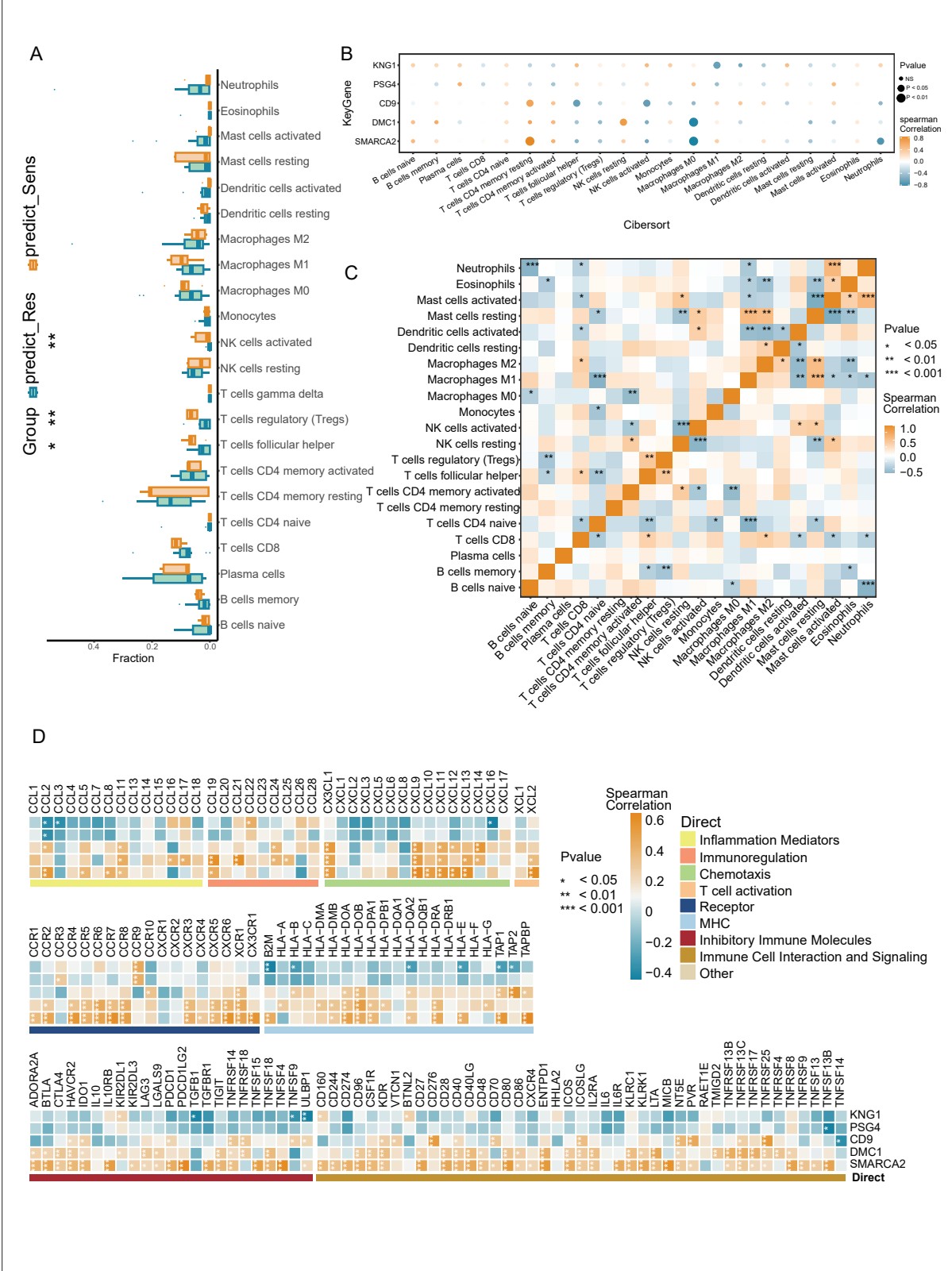

**Figure 3.** Annotation analysis of in-house NPC cohorts based on NPC-RSS predictive grouping with immune-related features. (**A**) Comparison of immune cell infiltration between NPC-sensitive and resistant tissue groups (n=34). *p<0.05, p<0.01, ***p<0.001. (**B**) Bubble plot depicting the correlation between the top 5 weight-ranked gene features in the NPC-RSS (SMARCA2, DMC1, CD9, PSG4, KNG1) and tumor-immune-infiltrating cells in the radiotherapy-sensitive group. Bubble size represents the proximity of the p-value to zero, with orange and blue colors indicating the strength of

*Figure 3 continued on next page*

*Figure 3 continued*

positive and negative correlations, respectively. (**C**) Analysis of interactions among 22 different immune cell types in patients from the NPC-sensitive group. *p<0.05, p<0.01, *p<0.001. (**D**) Correlation analysis of the top 5 weight-ranked genes in the NPC-RSS (SMARCA2, DMC1, CD9, PSG4, KNG1) with functionally diverse immune genes in the radiotherapy-sensitive group. *p<0.05, p<0.01, ***p<0.001.

profiles of cell-type-specific marker genes, we determined that these 17 cell subpopulations belonged to seven distinct cell types: T cells, myeloid cells, B cells, epithelial cells, neutrophils, fibroblasts, and mast cells (*Figure 4A*).

T cells expressed CD3D, CD8A, and CD4 genes. Myeloid cells expressed C1QB, LYZ, and AIF1 genes. B cells expressed MS4A1 and CD79A genes. Epithelial cells specifically expressed KRT5 and EPCAM genes. Neutrophils expressed specific marker genes. Fibroblasts expressed DCN and COL1A1 genes. Mast cells expressed CPA3, TPSAB1, and MS4A2 as marker genes. We used NPC-RSS to calculate the radiotherapy sensitivity scores of each cell. The predicted scores of cells in the sensitive group were significantly higher than those in the resistant group, especially in epithelial cells, myeloid cells, fibroblasts, and mast cells (*Figure 4B*). The sensitivity scores of all cells are shown in *Figure 4C*. Moreover, most of the 18 genes in the NPC-RSS were related to immune cells (*Figure 4D*). Interestingly, the radiotherapy-sensitive group showed significant dominance in terms of immune cells, whereas the radiotherapy-resistant group was mainly composed of malignant epithelial and stromal cells (*Figure 4E*). It is noteworthy that T cells, myeloid cells, epithelial cells, mast cells, and fibroblasts consisted of multiple cell subpopulations (*Figure 4F*), suggesting that these cell types may be heterogeneous.

## Pathway enrichment analysis findings

Next, we investigated the specific signaling pathways associated with the key genes of NPC-RSS and explored the potential molecular mechanisms by which these genes influence radiosensitivity in NPC. GSVA results demonstrated that high expression of SMARCA2 enriched signaling pathways such as WNT_BETA_CATENIN_SIGNALING, PI3K_AKT_MTOR_SIGNALING, TGF_BETA_SIGNALING, and NOTCH_SIGNALING (*Figure 5A*). Similarly, high expression of DMC1 enriched INTERFERON_GAMMA_RESPONSE and IL2_STAT5_SIGNALING pathways (*Figure 5B*).

Moreover, GSEA results showed that SMARCA2 was enriched in pathways such as the JAK-STAT signaling pathway, NF-kappa B signaling pathway, and T cell receptor signaling (*Figure 5C*). DMC1 was enriched in pathways such as cell adhesion molecules, Hedgehog signaling pathway, and phosphatidylinositol signaling system (*Figure 5D*). These findings suggest that NPC-RSS key genes may influence radiotherapy sensitivity in NPC patients through these pathways.

## Correlation between NPC-RSS key genes and radiosensitization genes

Radiosensitivity-related genes were obtained by searching the GeneCards database (https://www.genecards.org/) for the term "Radiosensitivity." The local NPC data were grouped and analyzed based on NPC-RSS. The expression levels of the top 20 genes, ranked by their Relevance scores, were correlated. The results, presented in *Figure 2—figure supplement 1*, showed significant differences in the expression of several radiosensitivity-related genes between the radiotherapy-resistant and sensitive groups (*Figure 5—figure supplement 1*).

Spearman correlation analysis indicated that the expression of SMARCA2 was positively correlated with high expression levels of TP53, EGFR, BRCA2, and ATM. Similarly, DMC1 expression was positively correlated with high expression levels of BRCA1, APTX, ATM, and CCR6, while CD9 expression was positively correlated with high expression levels of EGFR, TP53, ERBB2, and RAD51. The expression of KNG1 was correlated with high expression levels of HULC and significantly negatively correlated with high expression levels of BADA (*Figure 5E*).

## Discussion

NPC, a malignant tumor, is prevalent in East and Southeast Asia, particularly in South China. Radiotherapy is the primary treatment modality for NPC; however, treatment failure occurs in approximately 20–30% of patients (*Chua et al., 2016*). Radiotherapy sensitivity is a crucial factor influencing treatment outcomes in NPC patients. Thus, it is imperative to comprehensively investigate the mechanisms

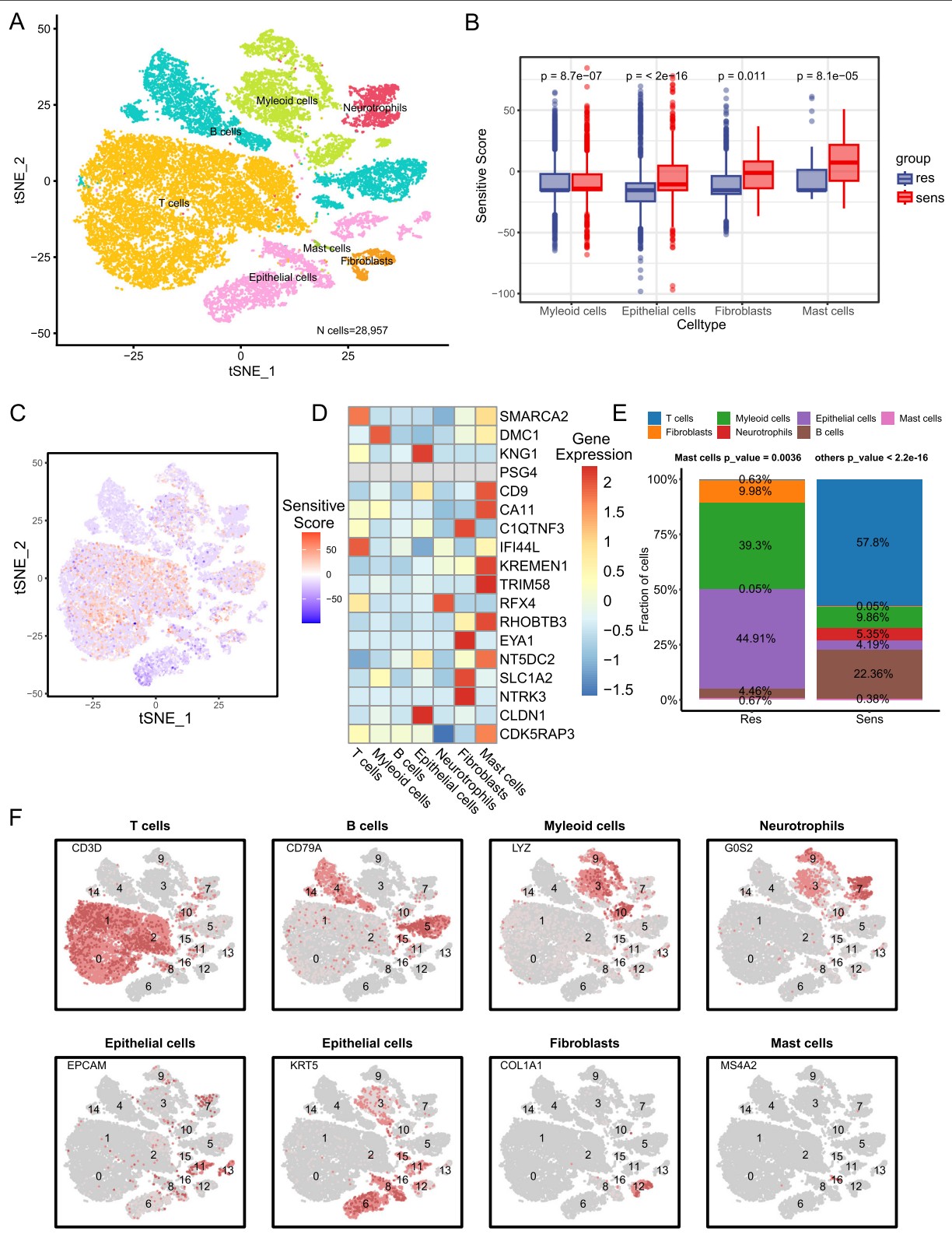

**Figure 4.** Biological characterization of key NPC-RSS genes at the single-cell level. (**A**) Clustered UMAP plot of three radiotherapy-sensitive and one radiotherapy-resistant sample with a total of 28,957 cells (n=4). Each color represents a cellular subpopulation (see cellular subpopulation annotations on the right). (**B**) Myeloid cells, epithelial cells, fibroblasts, and mast cells were significantly more abundant in samples from the radiotherapy-sensitive group compared to the radiotherapy-resistant group. (**C**) NPC-RSS scores displayed on all cells, with redder colors indicating higher scores. (**D**) NPC-RSS

*Figure 4 continued on next page*

*Figure 4 continued*

model gene expression in all cell subpopulations. Redder colors indicate higher expression, while bluer colors indicate lower expression. (**E**) Histogram showing the percentage of seven cell subpopulations in the radiotherapy-sensitive and radiotherapy-resistant groups. Different colors indicate different cell subpopulations. (**F**) Expression of marker genes used to annotate various cell subpopulations.

underlying radiotherapy sensitivity in NPC and identify effective therapeutic targets. Such investigations will facilitate the elucidation of therapeutic resistance mechanisms and provide a foundation for personalized treatment strategies. In the present study, we developed an integrative approach to establish a consistent nasopharyngeal carcinoma radiotherapy sensitivity signature (NPC-RSS) using NPC samples obtained from Zhujiang Hospital of Southern Medical University (*Figure 1*). In summary, we fitted 113 models to the training dataset using the leave-one-out cross-validation (LOOCV) framework. Further validation using the independent dataset GSE32389 from the Gene Expression Omnibus (GEO) database demonstrated that the glmBoost + NaiveBayes model exhibited the best performance.

The integrated procedure's advantage lies in its utilization of various machine learning algorithms and their combinations to fit models for NPC radiotherapy sensitivity, achieving a consensus performance. Furthermore, the combination of algorithms can reduce the dimensionality of the variables, resulting in a more simplified and easily translatable model. The predictive validity of NPC-RSS has been further validated using NPC cell lines generated at our local center. Moreover, we performed a related validation on local single-cell data, which revealed that the radiotherapy-sensitive group exhibited a significant predominance of immune cells, whereas the radiotherapy-resistant group primarily comprised malignant epithelial and stromal cells.

Previous studies have demonstrated that the core genes of NPC-RSS, including SMARCA2, DMC1, CD9, PSG4, and KNG1, are associated with tumor radiosensitization to varying degrees (*Zernickel et al., 2019*; *Xu et al., 2018b*; *Erovic et al., 2005*; *Jennrich et al., 2022*; *Xu et al., 2018a*). The protein encoded by the SMARCA2 gene plays a crucial role in various cellular processes, including cytokine response, DNA repair, regulation of cell proliferation, lineage specification and development, and cell adhesion (*Helming et al., 2014*; *Wang et al., 2014*; *Zhu et al., 2015*; *Tian et al., 2013*; *Dykhuizen et al., 2013*). These processes are intimately linked to the radiosensitivity of tumor cells. Cell proliferation and DNA repair are critical determinants of tumor cell response to radiotherapy, and the involvement of SMARCA2 may influence this response (*Lee et al., 2013*; *Su et al., 2019*; *Wang et al., 2019*; *Toulany et al., 2006*; *Hadjipanayis and DeLuca, 2005*). Furthermore, SMARCA2 is implicated in processes such as cell adhesion and cytokine response, which are tightly coupled to the interactions of tumor cells within the microenvironment. Cell adhesion influences the low-dose hyper-radiosensitivity of a broad spectrum of tumor cells, potentially further modulating their sensitivity to radiotherapy (*Mathur et al., 2023*).

Radiation is capable of inducing DNA double-strand breaks, and cells must repair this damage through DNA repair mechanisms to maintain genomic stability. DMC1, a key factor in homologous recombination, is involved in the repair process of DNA double-strand breaks (*Dunne et al., 2003*; *Zhang et al., 2017*; *Xie et al., 2019*). Consequently, abnormal function or altered expression levels of DMC1 may influence cellular sensitivity to radiation and, consequently, the efficacy of radiation therapy. CD9 is believed to play a role in various stages of cancer development (*Murayama et al., 2015*). As a cell surface protein, CD9 is involved in regulating cell-cell adhesion and interactions, which may affect the response of tumor cells to radiation therapy. Moreover, CD9 may impact tumor sensitivity to radiation therapy by modulating the metastatic and invasive properties of tumor cells (*Hori et al., 2004*; *Herr et al., 2013*; *Yin et al., 2014*; *Rappa et al., 2015*). It has been shown that all human PSGs initiate TGF-β1. In vivo, PSGs may increase the availability of active TGF-β1 in both soluble and matrix-bound forms of this potential cytokine (*Warren et al., 2018*). Furthermore, TGF-β1 levels accurately predict radiosensitization in elderly patients with unresectable NSCLC undergoing 3D-CRT (*Fu et al., 2014*).

Pregnancy-specific glycoproteins (PSGs) may influence radiosensitivity by regulating transforming growth factor-beta 1 (TGF-β1) activity. In radiation therapy, radiation can induce an inflammatory response in tissues and affect angiogenesis. Kininogen-1 (KNG1), a factor associated with coagulation and inflammatory response (*Abdullah-Soheimi et al., 2010*), may play a regulatory role in radio-sensitivity and the effects of radiation therapy. Furthermore, we found that the core genes of the

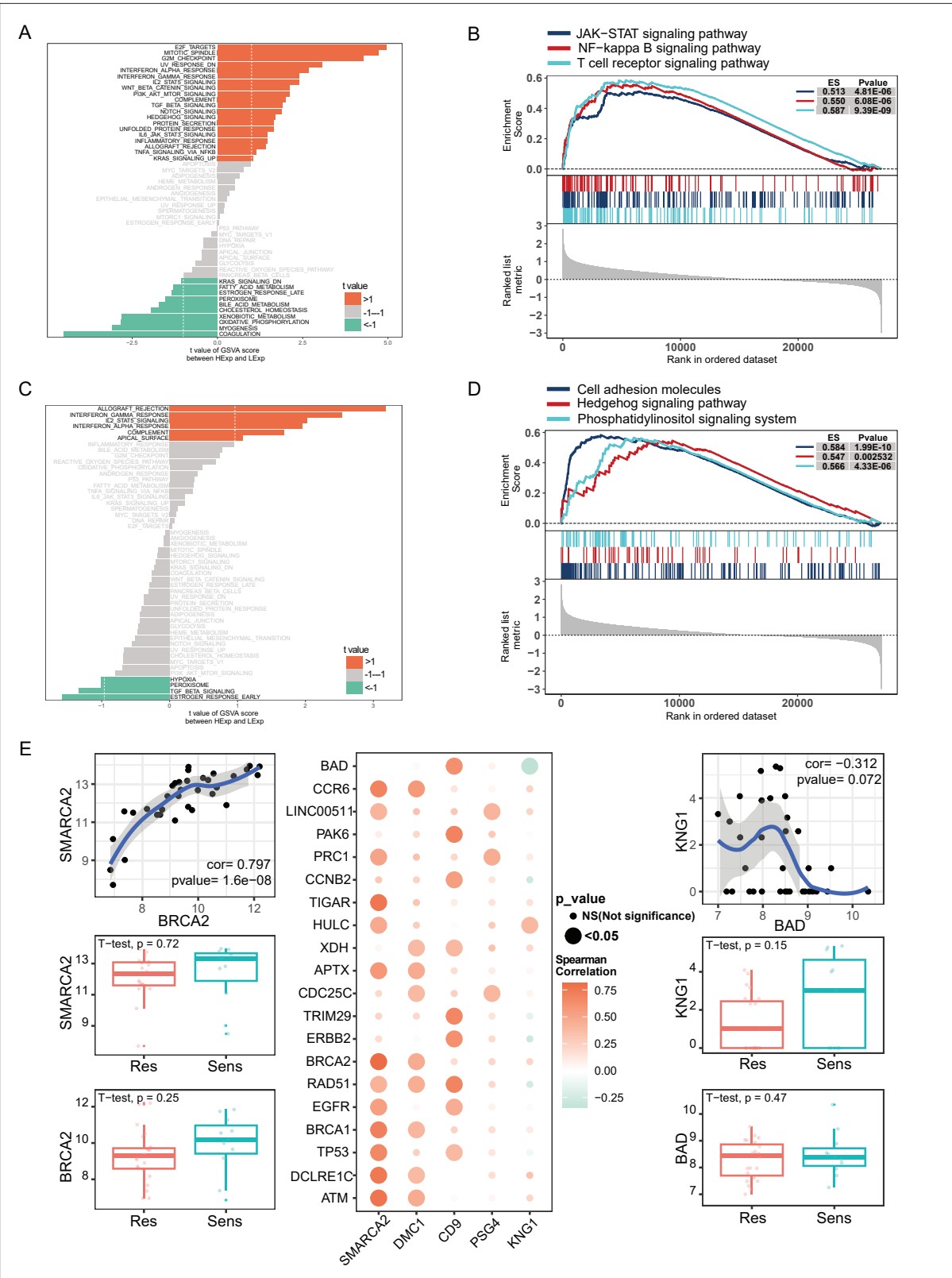

**Figure 5.** GSVA and GSEA analyses of NPC-RSS key genes and their correlation with radiosensitization genes. (**A**) GSVA of SMARCA2. (**B**) GSVA of DMC1. (**C**) GSEA of SMARCA2. (**D**) GSEA of DMC1. (**E**) Pearson correlation bubble plot of the top five NPC-RSS genes (SMARCA2, DMC1, CD9, PSG4, and KNG1) with radiosensitization-related genes. The larger the bubble, the closer the p-value is to zero; the more orange the color, the stronger the positive correlation; the more green the color, the stronger the negative correlation.

*Figure 5 continued on next page*

*Figure 5 continued*

The online version of this article includes the following figure supplement(s) for figure 5:

**Figure supplement 1.** Differences in the expression of radiotherapy sensitivity-related genes in NPC radiotherapy-sensitive and resistant groups.

nasopharyngeal carcinoma radiosensitivity signature (NPC-RSS), namely SMARCA2, DMC1, CD9, PSG4, and KNG1, had significant correlations with previously published genes related to radiosensitization (*Figure 5E*). The roles of these genes in regulating radiosensitization further support the importance and reliability of the NPC-RSS developed in this study for predicting radiosensitization in NPC.

Previous studies have shown that SMARCA2 has a significant effect on the responsiveness to radiotherapy in non-small cell lung cancer, and its depletion can increase the sensitivity to radiotherapy, suggesting that SMARCA2 may be a potential therapeutic target for sensitization to radiotherapy (*Zernickel et al., 2019*). In addition, the DMC1 gene was used in the construction of the radiosensitivity index (RSI) to assess the sensitivity and prognosis of radiotherapy, especially in endometrial cancer providing an important perspective on the DNA damage repair process (*Yang et al., 2024*). For CD9, its study in glioblastoma multiforme (GBM) revealed that the increase of CD9 and CD81 in extracellular vesicles (EVs) after radiotherapy was closely related to the cytotoxic response after treatment, which provided a new biomarker for radiotherapy monitoring (*Jennrich et al., 2022*). In contrast, the relationship between PSG4 and KNG1 and radiotherapy resistance has not been reported. In the future, the expression pattern of SMARCA2 in NPC patients and its relationship with the effect of radiotherapy can be analyzed by clinical samples to determine its potential as a therapeutic target for sensitization to radiotherapy. For DMC1, studying the correlation between its expression level and radiotherapy sensitivity in NPC may be helpful in predicting the efficacy and adjusting the treatment regimen. Meanwhile, by analyzing EVs, especially CD9-containing vesicles, in NPC patients after radiotherapy, we assessed its feasibility as a biomarker to monitor radiotherapy response. These studies not only contribute to an in-depth understanding of the mechanism of these genes in NPC radiotherapy, but may also open up new strategies to improve the efficacy of radiotherapy.

Radiotherapy plays a pivotal role in cancer treatment, and there is growing evidence of the potential for synergistic effects between radiation therapy and components of the immune response (*Ngwa, 2018*; *Binder et al., 2015*). However, there are limited data depicting the tumor immune microenvironment (TIME) associated with radiotherapy sensitivity. Indeed, recent preclinical studies have demonstrated that irradiated tumors can act as adjuvants to the immune system, promoting local tumor control and inducing regression of distant tumor deposits through abscopal effects (*Ngwa et al., 2018*). In addition to potential systemic effects, radiation significantly alters the distribution of tumor antigens, recruits anti-tumor T cells, and induces a distinct inflammatory microenvironment (*Demaria et al., 2021*). Some studies have shown that tumors with high immune cell infiltration are more sensitive to radiation (*Grass et al., 2022*); however, the relationship between radiation and immune response remains complex and context-dependent, and the optimal approach to integrating radiation and immunotherapy remains uncertain.

Studies have demonstrated that over one-third of tumor types in the radiotherapy-sensitive group of solid tumors exhibit an enrichment of CD8 + T cells, M1 macrophages, and NK + cells (*Grass et al., 2022*). To further investigate this phenomenon, we analyzed the immune infiltration between different subgroups of NPC-RSS and annotated the key NPC-RSS genes with immune-related features. The results demonstrated that the NPC radiosensitized group exhibited a higher level of immune infiltration and immune initiation, characterized by a greater abundance of activated NK cells, regulatory T cells (Tregs), and follicular helper T cells. The NPC-RSS key genes also exhibited a significant correlation and synergism with immune-related features. Notably, this synergistic correlation was most prominent in chemokines and chemokine receptor families, the major histocompatibility complex (MHC), immune cell interactions, and signaling pathways. These findings suggest that the radiosensitivity estimated by the NPC-RSS may be associated with a distinct tumor immune microenvironment, which could potentially be exploited through a combinatorial immunotherapy approach.

Notably, the results obtained from the collected single-cell data were consistent with the predicted trends of our constructed model. The radiotherapy-sensitive group exhibited a significant predominance of immune cells, while the radiotherapy-resistant group primarily comprised malignant epithelial and stromal cells. Furthermore, utilizing NPC-RSS to calculate the radiotherapy sensitivity score for each cell revealed that the predicted scores of cells in the sensitized group were significantly

higher than those in the resistant group, particularly in epithelial cells, myeloid cells, fibroblasts, and mast cells. This finding suggests that the presence of immune cells might have influenced the tumor's responsiveness to treatment to some extent prior to radiation therapy. Tumors with a predominance of immune cells may be more likely to exhibit a positive response to radiation therapy, either due to the involvement of immune cells in tumor surveillance and clearance or because the responsiveness of tumors to radiation therapy is modulated by immune cells.

Furthermore, pathway enrichment analysis suggested that the Wnt/β-catenin signaling pathway may be associated with the radiosensitivity of NPC. Gene Set Variation Analysis (GSVA) and Gene Set Enrichment Analysis (GSEA) indicated that the differential expression levels of SMARCA2, DMC1, CD9, PSG4, and KNG1 in the core of the NPC radiosensitivity signature (NPC-RSS) might influence various signaling pathways associated with disease progression, involving multiple biological processes such as DNA damage repair, cell death, immune regulation, cell proliferation, metabolism, tumor micro-environment regulation, and cell survival. For instance, the SMARCA2 high-expression group, which was the most significant, was enriched in the UV response, Wnt/β-catenin signaling pathway, NOTCH signaling pathway, TGFβ signaling pathway, PI3K-AKT-mTOR signaling pathway, Interferon Alpha response, Interferon Gamma response, and other signaling pathways. Previous studies have demonstrated that proteins related to the Wnt/β-catenin signaling pathway (DKK-3, β-catenin, and c-MYC) are involved in the development of NPC, and that knockdown of FOXO3a promotes radioresistance in NPC through the Wnt/β-catenin signaling pathway (*Pang et al., 2019*; *Luo et al., 2019*). Therefore, we hypothesized that SMARCA2 may regulate the radiosensitivity of NPC via the Wnt/β-catenin signaling pathway.

Incorporating the NPC-RSS scoring system into clinical decision-making and patient management involves several key steps: first, making genetic testing a standard part of nasopharyngeal cancer diagnosis and ensuring that physicians have timely access to scoring results to guide treatment planning. Second, physicians need to use the scoring results to customize individualized treatment plans and discuss them in a multidisciplinary team to optimize decision-making. At the same time, physicians should explain the clinical significance of the scores in detail and communicate effectively with patients to achieve shared decision-making. In addition, continuously monitoring the relationship between scoring and treatment outcomes, optimizing the scoring model based on actual data, and ensuring the integration of technology platforms and regulatory compliance are essential to safeguard the effective operation of the scoring system and the security of patient information.

The novelty of this study lies in the utilization of NPC expression data collected from Zhujiang Hospital of Southern Medical University, combined with multiple machine learning models, to develop NPC-RSS, a tool capable of effectively identifying the patient population more sensitive to radiotherapy. The tool was validated using single-cell data and cell lines from the in-house cohort, with the validation results aligning with the predictions of our constructed model. However, this study has several limitations. First, this study focuses on specific types of NPC, and thus may not be directly generalizable to other subtypes or related head and neck cancers. Second, the sample size of the dataset is small, which may affect the generalization and extrapolation ability of the model. To minimize the impact of the small sample size, we used advanced statistical techniques, such as cross-validation, to enhance the stability and credibility of the results. Nonetheless, we realize that larger datasets are necessary, so we are collaborating with other research centers to expand our sample size to make our findings more robust and widely applicable. Further, while our study relies on bioinformatics methods to identify and analyze key genes, the lack of direct experimental validation of function is a glaring deficiency. We are currently seeking additional funding support and plan to collaborate with laboratories to conduct the necessary functional validation experiments, which will help to further confirm the specific role of these genes in radiotherapy. In addition, we recognize that there may be a risk of overfitting when using machine learning algorithms to process biomedical data. For this reason, we not only implemented regularization techniques during the model development stage, but also adopted a rigorous cross-validation strategy during model validation. These measures help ensure that our models maintain good predictive performance on unseen data.

Despite these limitations, our study provides new insights into understanding the molecular mechanisms of radiotherapy sensitivity and points to possible directions for future research. Future work will focus on expanding the dataset, performing experimental validation, and further optimizing our predictive model.

# Materials and methods

## NPC patient data sources and data processing

We downloaded chemotherapy-related gene expression data from the GEO database for 20 NPC patients (GSE32389). Additionally, we collected samples from 34 NPC patients treated at Zhujiang Hospital of Southern Medical University between March 2017 and July 2021 (in-house cohort). The clinical characteristics of the NPC patients in the in-house cohort are detailed in *Supplementary file 1*. Written informed consent was obtained from the NPC patients whose NPC tissues were collected, and the study was approved by the Ethics Committee of Zhujiang Hospital of Southern Medical University. In this study, the in-house cohort was used as the training set, and GSE32389 was used as the validation set. The raw transcriptome sequencing data from the in-house cohort underwent quality control and preprocessing, which included data quality assessment, removal of low-quality reads, and removal of adapter or primer sequences using tools such as Trimmomatic and FastQC. Subsequently, the pre-processed reads were aligned to a reference genome or transcriptome using the alignment software HISAT2 to identify and localize RNA sequences. The aligned reads were then quantified using featureCounts to obtain the number of reads per gene or transcript. Gene expression levels were then quantified by normalizing the read counts to the length of each gene and the total number of reads, calculating the Fragments Per Kilobase Million (FPKM) values. The data for the GEO database were obtained from the Affymetrix GPL570 platform. Microarray data from the Affymetrix platform were background corrected, quantile normalized, and log2 transformed using the R package 'affy' with the RMA algorithm (*Irizarry et al., 2003*).

## Construction of NPC-RSS

In the training set, we performed differential analysis of the gene expression data between nasopharyngeal carcinoma patients who were sensitive or resistant to radiotherapy using the limma R package. This analysis identified 71 DEGs (p<0.05 and |LogFC|>0.585, *Supplementary file 2*; *Liu et al., 2015*). Next, we used these DEGs to develop a binary classification model using machine learning algorithms to predict radiotherapy sensitivity in nasopharyngeal carcinoma patients. In the data preprocessing stage, we performed robust multi-array average (RMA) normalization on the gene expression data to ensure that it met the input requirements of the machine learning algorithms, which assume a normal distribution.

First, we integrated a combination of 12 machine learning algorithms, including Least Absolute Shrinkage and Selection Operator (LASSO), Ridge, Elastic Net (Enet), stepwise generalized linear models (Stepglm), support vector machines (SVM), gradient boosting with component-wise linear models (glmBoost), linear discriminant analysis (LDA), partial least squares generalized linear models (plsRglm), random forest (RF), gradient boosting machine (GBM), extreme gradient boosting (XGBoost), and naive Bayes. Among these, RF, LASSO, glmBoost, Stepglm with both forward and backward selection (Stepglm[both]), and Stepglm with backward selection (Stepglm[backward]) can perform feature selection and dimensionality reduction. We combined these algorithms with the others to form 113 combinations of machine learning algorithms. In each of these 113 combinations, one algorithm was used for feature selection, while the other was used to construct a binary classification model. The performance of these models was evaluated using cross-validation to ensure robustness. (2) Next, in the training set, we used these 113 combinations to construct predictive models using the expression data of the 71 DEGs. A total of 48 models were successfully constructed. (3) In the validation set, we used these predictive models to calculate a radiotherapy sensitivity score for each NPC patient. (4) We evaluated the predictive performance of these models in the training and validation sets using ROC curves and the average AUC. The final model was named the Nasopharyngeal Carcinoma Radiosensitivity Score (NPC-RSS).

## Validating key genes of NPC-RSS at the cellular level

### Cell culture

The CNE2 nasopharyngeal carcinoma cell line, previously preserved by the Pearl River Hospital Center for Translational Medicine, was cultured in RPMI 1640 medium supplemented with 10% fetal bovine serum and 1% penicillin/streptomycin. The cells were maintained in a humidified incubator at 37 °C with 5% $CO_2$. Cell passaging and division ratios were carefully monitored and maintained

within optimal ranges to ensure cell health and experimental consistency. Regular cell line authentication was performed to ensure the use of the correct cell line. Stringent quality control measures were implemented for the culture medium and fetal bovine serum to minimize experimental variability. Precise addition of penicillin/streptomycin was ensured to maintain bacterial resistance while minimizing potential adverse effects on cell growth and viability. Appropriate cryopreservation and thawing protocols were followed to ensure optimal cell preservation and recovery. Strict adherence to aseptic techniques was maintained throughout the cell culture process to prevent bacterial and fungal contamination. Continuous monitoring of the incubator's internal environment was performed to maintain stable $CO_2$ concentration and temperature. Meticulous execution of these steps is crucial for maintaining the stability and reliability of the cell culture system.

## Cell irradiation and construction of CNE2-RS radiation-resistant cells

A 6 MV photon beam was generated at room temperature using a linear accelerator (Elekta, SYNERGY PL, Switzerland) and delivered to the cells at a 180° gantry angle with a dose rate of approximately 600 cGy/min. The bottom of the cell culture vessel was covered with a 1.5-cm-thick tissue-equivalent bolus. The source-to-surface distance (SSD) was 100 cm.

Each T25 culture flask was seeded with approximately 2x105 cells in the exponential growth phase and sequentially irradiated with increasing doses (2, 4, 6, 8, and 10 Gy) twice, followed by a single 10 Gy dose, for a total of nine irradiations and a cumulative dose of 50 Gy. Following the initial 2 Gy irradiation, the cells were cultured and passaged twice before being irradiated with another 2 Gy. The surviving cells were then subjected to the aforementioned irradiation schedule, and the resulting population was designated as the radiation-resistant cell line CNE-2-RS (*Figure 2E*). Parental cells were used as controls and underwent the same treatment regimen without irradiation.

To maintain radioresistance, the CNE-2-RS cells used in the experiments were exposed to a low dose of approximately 1–2 Gy per month. Cells within 15 passages following the low-dose irradiation were used for subsequent experiments, and all experiments were conducted using cells in the exponential growth phase. The radioresistance of CNE-2-RS cells was subsequently validated using clonogenic assays, with results consistently confirming the maintained radioresistant phenotype.

## RT-qPCR

Total RNA was extracted from CNE2 cells using TRIzol reagent (Invitrogen). Following the measurement of RNA concentration with a Nanodrop 2000 instrument (Thermo Scientific), cDNA synthesis was performed using the Color Reverse Transcription Kit (EZBioscience, USA). The relative mRNA expression levels of each gene were calculated using the 2^(-ΔΔCT) method. The primer sequences of the genes are presented in *Supplementary file 3*.

## Annotation of immune-related features of NPC-RSS key genes

The CIBERSORT algorithm was applied to analyze the expression data of nasopharyngeal cancer patients to infer the relative proportions of 22 immune-infiltrating cells. Additionally, a correlation analysis was performed between gene expression and immune cell content using Spearman's method. Results with a p-value <0.05 were considered statistically significant. After grouping the nasopharyngeal cancer data based on the previously extracted NPC-RSS, the CIBERSORT method was applied to quantify the immune cell infiltration. The objective was to thoroughly investigate whether there was a significant difference in immune infiltration between the sensitive and resistant groups after stratifying nasopharyngeal cancer patients according to their NPC-RSS. To further elucidate the potential association of immune features in nasopharyngeal cancer, we also investigated the correlation of NPC-RSS key genes with immune-related feature genes obtained from the TISIDB database, including chemokine-related genes, immunosuppressant-related genes, MHC-related genes, immune stimulation-related genes, and receptor-related genes (*Ru et al., 2019*).

## Further validation of NPC-RSS at the single-cell level

The single-cell RNA sequencing (scRNA-seq) data were processed and quantified using the Cell Ranger (version 2.0.1) pipeline. Initially, reads obtained from 10 x Genomics were aligned to the human reference genome hg19. The FASTQ sequence files of four samples were aligned to the human reference genome hg19 using STAR software and the Cell Ranger 'count' module, generating files containing

barcode tables and gene tables. The raw gene expression matrices for each sample, generated by the CellRanger software, were merged in R and converted to Seurat objects for downstream analysis using the Seurat R package. Subsequently, cells with the following characteristics were removed: (1) number of unique molecular identifiers (UMIs) less than 201, (2) fewer than 100 or more than 6000 expressed genes (potential cellular duplicates), and (3) percentage of mitochondrial genes greater than 20%. Additionally, genes expressed in fewer than three cells were removed from the dataset. After quality control, 28,957 high-quality NPC cells were retained for further analysis. Subsequently, the gene expression matrix was normalized to the total cell reads and mitochondrial reads using linear regression with Seurat's RegressOut function. Following data normalization, highly variable genes were identified across all individual cells while controlling for the relationship between mean expression and dispersion. Prior to the joint analysis of the four-sample scRNA-seq dataset, we employed the R package Harmony to mitigate batch effects and minimize their impact on downstream analyses (*Korsunsky et al., 2019*). Furthermore, we utilized Seurat's 'CellCycleScoring' function to identify the expression of cell cycle-related genes across various cell subpopulations. We employed a previously defined core set of 43 G1/S and 54 G2/M cell cycle genes (*Tirosh et al., 2016*). Cells were categorized based on the maximum average expression ('cycle score') across both gene sets. Following cell cycle analysis, no bias attributable to cell cycle genes was observed.

## Cell clustering analysis and identification of marker genes for cellular subpopulations

Normalization, clustering, differential gene expression analysis, and visualization were performed using the Seurat 3.1 R package. To reduce the dimensionality of the dataset, principal component analysis (PCA) was performed on the 2000 variably expressed genes. The top 20 principal components were then used as input for the t-distributed stochastic neighbor embedding (t-SNE) algorithm, which was run using the default settings of the RunTSNE function in Seurat. Cell clustering was performed using the 'FindClusters' function in Seurat with a resolution of 0.3, resulting in the identification of 17 distinct cell clusters (clusters 0–16). Subsequently, the FindMarkers function was employed to identify specific marker genes for each cell subpopulation. The process was conducted as follows: (1) DEGs for each cell subpopulation were identified by comparing the cells of that subpopulation to all other cells using the Wilcoxon rank-sum test; (2) Bonferroni-corrected p-values less than 0.05 were used as the cut-off for determining statistically significant DEGs; (3) Genes with an average expression level more than twofold higher in a specific cell subpopulation compared to other subpopulations were selected as marker genes. Cell type annotation for each subpopulation was performed based on marker genes reported in previous studies (*Chen et al., 2020*), the CellMarker database, and the R package 'SingleR' (*Aran et al., 2019*).

## GSVA and GSEA analysis of NPC-RSS key genes

GSVA and GSEA analysis of key NPC-RSS genes Based on the expression levels of key NPC-RSS genes, we stratified the patients into high and low expression groups and thoroughly investigated the signaling pathway differences between these groups using Gene Set Enrichment Analysis (GSEA). We utilized the Molecular Signatures Database (MSigDB) gene sets as the annotated gene sets for the subtype pathways to conduct differential expression analysis between subtypes. Gene consistency was assessed using a concordance score calculated by sorting the genes according to their LogFC (adjusted p<0.05). Furthermore, to investigate the association of NPC-RSS with known radiosensitivity-related genes, we searched for 'Radiosensitivity' in the GeneCards database (https://www.genecards.org/) to obtain radiosensitization-related genes and examined the expression levels of the top 20 genes based on their Relevance scores and their correlation with the top-5 genes in NPC-RSS.

### Statistical analysis

The differentially expressed genes between NPCs of the sensitive and resistant groups in the dataset were identified using the R package 'limma'. For normally distributed variables, statistical differences between two groups were determined by two-tailed t-test, while one-way ANOVA was used for multiple group comparisons. For non-normally distributed variables, statistical differences between two groups were determined by the Wilcoxon rank-sum test, while the Kruskal-Wallis test was used for multiple group comparisons. All statistical analyses were conducted using R software (version 4.1.2).

Each experiment was performed independently in triplicate unless otherwise specified. Spearman's rank correlation analysis was employed to assess the correlation between two variables.

In the single-cell sequencing data processing section, comparisons between two groups were performed using the unpaired two-tailed t-test (implemented in the R package limma) or the Mann-Whitney U test. All statistical analyses and graph visualizations were conducted using the R programming language.

For all experiments, samples from the same patient were processed in parallel. Cells from each patient sample were subjected to single-cell RNA sequencing (10 x Genomics) simultaneously but in different passages and vials. Boxplots were generated using the R base software package with default parameters. The boxes span the interquartile range (IQR; from the 25th to the 75th percentile), with the center line representing the median. The lower whiskers extend to the smallest value no further than 1.5×IQR from the 25th percentile. The upper whiskers extend to the largest value no further than 1.5×IQR from the 75th percentile. Due to the large number of data points represented in the box-and-whisker plots, individual data points were not shown to maintain clarity in the overall distribution.

## Conclusion

In this study, we developed a robust NPC-RSS using multiple bioinformatics and machine learning algorithms. The expression of key genes SMARCA2, DMC1, CD9, and KNG1 was validated to be consistent with the NPC-RSS in our in-house cell lines. Compared to the resistant group, the sensitive group classified by NPC-RSS exhibited a more abundant and active state of immune infiltration. Furthermore, in single-cell samples, NPC-RSS was elevated in the radiotherapy-sensitive group, where immune cells played a predominant role. Additional analysis uncovered substantial correlations between key NPC-RSS genes and genes previously demonstrated to be involved in the regulation of radiosensitization. Single-gene pathway enrichment analysis revealed that the top 2 NPC-RSS genes were associated with several signaling pathways, including the Wnt/β-catenin signaling pathway, JAK-STAT signaling pathway, NF-kappa B signaling pathway, and T cell receptor signaling pathway. These pathways may influence the immune microenvironment of NPC, thereby affecting its susceptibility to radiotherapy. Our findings provide a novel perspective for understanding NPC radiosensitivity.

## Acknowledgements

This work was supported by grants from the Natural Science Foundation of Guangdong Province (2021A1515012593), the National Natural Science Foundation of China (82373129, 82172750, 82172811 and 82260546), the Guangdong Basic and Applied Basic Research Foundation (Guangdong-Guangzhou Joint Funds) (2022A1515111212), the Science and Technology Program of Guangzhou City (2023A04J1257).

## Additional information

### Funding

| Funder | Grant reference number | Author |
| --- | --- | --- |
| Natural Science Foundation of Guangdong Province | 2021A1515012593 | Jian Zhang |
| The National Natural Science Foundation of China | 82172811 | Jian Zhang |
| Guangdong Basic and Applied Basic Research Foundation | 2022A1515111212 | Jian Zhang |
| Science and Technology Program of Guangzhou City | 2023A04J1257 | Jian Zhang |

| Funder | Grant reference number | Author |
|---|---|---|

The funders had no role in study design, data collection and interpretation, or the decision to submit the work for publication.

## Author contributions

Kailai Li, Data curation, Formal analysis, Validation, Investigation, Visualization, Methodology, Writing – original draft; Junyi Liang, Visualization, Writing – original draft; Nan Li, Data curation, Methodology, Writing – original draft; Jianbo Fang, Data curation, Writing – original draft; Xinyi Zhou, Jian Zhang, Anqi Lin, Methodology, Writing – original draft; Peng Luo, Formal analysis, Funding acquisition, Methodology, Writing – original draft; Hui Meng, Formal analysis, Funding acquisition, Writing – original draft

## Author ORCIDs

Kailai Li ⓘ https://orcid.org/0009-0000-7444-995X
Jian Zhang ⓘ https://orcid.org/0000-0001-7217-0111
Peng Luo ⓘ https://orcid.org/0000-0002-8215-2045

## Ethics

This study analyzed publicly available human data from the GEO database (GSE32389). The original data was collected with proper ethical approval and patient consent as described in the original publication (Yang S et al., Int J Oncol 2012).

Reviewer #1 (Public review): https://doi.org/10.7554/eLife.99849.3.sa1
Reviewer #2 (Public review): https://doi.org/10.7554/eLife.99849.3.sa2
Author response https://doi.org/10.7554/eLife.99849.3.sa3

---

# Additional files

## Supplementary files

Supplementary file 1. Baseline characteristics of the 34 NPC samples.

Supplementary file 2. Baseline characteristics of the 4 NPC samples.

Supplementary file 3. Results of differential analysis of differential genes for which machine learning was performed.

Supplementary file 4. Analysis software and pipeline description.

MDAR checklist

## Data availability

The publicly available dataset GSE32389 was obtained from the Gene Expression Omnibus (GEO) database. All source code used for data analysis is available on GitHub, copy archived at *Li, 2025*. The raw and processed data used in this study, including source data for all figures and tables, are publicly available on Mendeley Data. The analysis pipeline and workflow are described in detail in *Supplementary file 4*.

The following dataset was generated:

| Author(s) | Year | Dataset title | Dataset URL | Database and Identifier |
|---|---|---|---|---|
| Li K | 2025 | A multi-gene predictive model for the radiation sensitivity of nasopharyngeal carcinoma based on machine learning | https://doi.org/10.17632/zc6k6d5c9k.1 | Mendeley Data, 10.17632/zc6k6d5c9k.1 |

The following previously published dataset was used:

| Author(s) | Year | Dataset title | Dataset URL | Database and Identifier |
|---|---|---|---|---|
| Yang S, Chen J, Guo Y, Zhang Z, Chen K, Wu H, Li Y | 2013 | Nasopharyngeal carcinoma patient samples: radio-sensitive samples vs. radio-resistant samples | http://www.ncbi.nlm.nih.gov/geo/query/acc.cgi?acc=GSE32389 | NCBI Gene Expression Omnibus, GSE32389 |

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
