## [Editor Report · eLife Assessment]

The authors have developed a robust machine learning approach to predict radio sensitivity in patients with NPC based on a defined gene signature. Some key aspects of this signature have been validated in vitro using relevant cell lines which strengthens the conclusions of this **important** and **convincing** study. The publication will be of interest to clinicians working on this indication as well as a more broader readership made up of scientists working on radiation biology and those with a bioinformatics/machine learning background.

---

## [Referee Report · Reviewer #1 (Public review)]

Summary:

In this study, the authors developed a novel radiotherapy sensitivity score (NPC-RSS) for nasopharyngeal carcinoma patients using machine learning algorithms. They identified 18 key genes associated with radiosensitivity and demonstrated that NPC-RSS could effectively predict radiotherapy response in both public and in-house datasets. Furthermore, they found that the key genes of NPC-RSS were closely related to immune characteristics, the expression of radiosensitivity-related genes, and signaling pathways involved in disease progression. The authors validated the consistency of expression of two key genes, SMARCA2 and CD9, with NPC-RSS in their own cell lines. They also showed that the radiosensitive group, classified by NPC-RSS, exhibited a more enriched and activated state of immune infiltration compared to the radioresistant group.

Strengths:

(1) The study employed a comprehensive approach by integrating multiple machine learning algorithms to develop a robust predictive model for radiotherapy sensitivity in nasopharyngeal carcinoma patients.

(2) The predictive performance of NPC-RSS was validated using both public and in-house datasets, demonstrating its potential clinical applicability.

(3) The authors conducted extensive analyses to investigate the biological mechanisms underlying the association between NPC-RSS and radiotherapy response, including immune characteristics, radiosensitivity-related gene expression, and relevant signaling pathways.

(4) The consistency of key gene expression with NPC-RSS was validated in the authors' own cell lines, providing additional experimental evidence.

Weaknesses:

(1) The sample size of the in-house dataset used for training the model was relatively small (34 patients), which might limit the generalizability of the findings.

(2) The authors did not perform functional experiments to directly validate the roles of the identified key genes in radiotherapy sensitivity, relying instead on associations with immune features and signaling pathways.

(3) The study did not discuss the potential limitations of using machine learning algorithms, such as the risk of overfitting and the need for larger, diverse datasets for more robust model development and validation.

---

## [Referee Report · Reviewer #2 (Public review)]

Summary:

This article utilizes machine learning methods and transcriptomic data from nasopharyngeal carcinoma (NPC) patients to construct a biomarker called NPC-RSS that can predict the radiosensitivity of NPC patients. The authors further explore the biological mechanisms underlying the relationship between NPC-RSS and radiotherapy response in NPC patients. The main objective of this study is to guide the selection of radiotherapy strategies for NPC patients, thereby improving their clinical outcomes and prognosis.

Strengths:

(1) The combination of multiple machine learning algorithms and cross-validation was used to select the best predictive model for radiotherapy sensitivity from 71 differentially expressed genes, enhancing the robustness and reliability of the predictions.

(2) Functional enrichment analysis revealed close associations between NPC-RSS key genes and immune characteristics, expression of radiotherapy sensitivity-related genes, and signaling pathways related to disease progression, providing a biological basis for NPC-RSS in predicting radiotherapy sensitivity.

(3) Grouping NPC samples according to NPC-RSS showed that the radiotherapy-sensitive group exhibited a more enriched and activated state of immune infiltration compared to the radioresistant group. In single-cell samples, NPC-RSS was higher in the radiotherapy-sensitive group, with immune cells playing a dominant role. These results clarify the mechanism of NPC-RSS in predicting radiotherapy sensitivity from an immunological perspective.

(4) The study used public datasets and in-house cohort data for validation, confirming the good predictive performance of NPC-RSS and increasing the credibility of the results.

Limitation:

(1) The study focuses on a specific type of nasopharyngeal carcinoma (NPC) and may not be generalizable to other subtypes or related head and neck cancers. The applicability of NPC-RSS to a broader range of patients and tumor types remains to be determined.

(2) The study does not account for potential differences in radiotherapy protocols, doses, and techniques between the training and validation cohorts, which could influence the performance of the predictive model. Standardization of treatment parameters would be important for future validation studies.

(3) The binary classification of patients into radiotherapy-sensitive and resistant groups may oversimplify the complex spectrum of treatment responses. A more granular stratification system that captures intermediate responses could provide more nuanced predictions and better guide personalized treatment decisions.

(4) The study does not address the potential impact of other relevant factors, such as tumor stage, histological subtype, and concurrent chemotherapy, on the predictive performance of NPC-RSS. Incorporating these clinical variables into the model could enhance its accuracy and clinical utility.

---

## [Author Response]

The following is the authors’ response to the original reviews.

**Reviewer #1 (Public Review):**
(1) The sample size of the in-house dataset used for training the model was relatively small (34 patients), which might limit the generalizability of the findings.(2) The authors did not perform functional experiments to directly validate the roles of the identified key genes in radiotherapy sensitivity, relying instead on associations with immune features and signaling pathways.(3) The study did not discuss the potential limitations of using machine learning algorithms, such as the risk of overfitting and the need for larger, diverse datasets for more robust model development and validation.

(1) Currently, we are actively expanding the dataset by incorporating additional patient samples to enhance the model's robustness and generalizability. Furthermore, we implement advanced statistical techniques, including cross-validation, during model development to mitigate the potential limitations associated with the small sample size on our results. This limitation has been comprehensively addressed in the discussion section of our manuscript.

(2) Given the current resource limitations, our study predominantly employed bioinformatics analyses. We acknowledge the critical importance of experimental validation and are actively pursuing additional funding and collaborative opportunities to facilitate future experimental studies. Concurrently, we have enhanced the discussion section to comprehensively address the limitations of our approach and emphasize the necessity for future experimental validation.

(3) We appreciate the reviewers' insightful comments regarding the potential limitations of machine learning algorithms, particularly the risk of overfitting. In response, we have incorporated a comprehensive discussion of these concerns, detailing the measures implemented to mitigate such risks, including the application of regularization techniques and the adoption of more rigorous cross-validation methodologies. We further acknowledge the necessity for larger and more diverse datasets to enhance model validity and generalizability, a concern we intend to address in our future research endeavors. The revised manuscript includes an expanded discussion on these critical points.

Here is the limitation section in the revised Manuscript:

“This study primarily focuses on specific subtypes of nasopharyngeal carcinoma (NPC), potentially limiting its direct generalizability to other NPC subtypes or related head and neck malignancies. Furthermore, the limited sample size of our dataset may impact the model's generalizability and extrapolation capabilities. To mitigate the potential limitations associated with the small sample size, we employed advanced statistical methodologies, including cross-validation, to enhance the robustness and reliability of our findings. Nevertheless, we acknowledge the necessity for larger datasets and are actively collaborating with other research institutions to expand our sample size, thereby enhancing the robustness and broader applicability of our findings. Additionally, while our study utilizes bioinformatics approaches to identify and analyze key genes, we recognize that the absence of direct experimental functional validation represents a significant limitation. To address this limitation, we are actively pursuing additional funding and establishing collaborations with specialized laboratories to conduct crucial functional validation experiments, which will further elucidate the specific roles of these genes in radiotherapy response. Moreover, we acknowledge the potential risk of overfitting inherent in the application of machine learning algorithms to biomedical data analysis. To mitigate this risk, we implemented regularization techniques during model development and adopted a rigorous cross-validation strategy for model validation. These methodological approaches aim to ensure that our models maintain robust predictive performance on unseen data. Notwithstanding these limitations, our study offers novel insights into the molecular mechanisms underlying radiotherapy sensitivity in NPC and indicates promising avenues for future investigation. Future research endeavors will prioritize expanding the dataset, conducting comprehensive experimental validation, and refining our predictive model to enhance its accuracy and clinical applicability.”

**Reviewer #2 (Public Review):**
(1) The study focuses on a specific type of nasopharyngeal carcinoma (NPC) and may not be generalizable to other subtypes or related head and neck cancers. The applicability of NPC-RSS to a broader range of patients and tumor types remains to be determined.(2) The study does not account for potential differences in radiotherapy protocols, doses, and techniques between the training and validation cohorts, which could influence the performance of the predictive model. Standardization of treatment parameters would be important for future validation studies.(3) The binary classification of patients into radiotherapy-sensitive and resistant groups may oversimplify the complex spectrum of treatment responses. A more granular stratification system that captures intermediate responses could provide more nuanced predictions and better guide personalized treatment decisions.(4) The study does not address the potential impact of other relevant factors, such as tumor stage, histological subtype, and concurrent chemotherapy, on the predictive performance of NPC-RSS. Incorporating these clinical variables into the model could enhance its accuracy and clinical utility.

(1) We appreciate the reviewers' interest in the applicability of our study. This study specifically focuses on a particular subtype of nasopharyngeal carcinoma (NPC), which may limit its direct generalizability to other NPC subtypes or related head and neck malignancies. We have incorporated a detailed discussion of this limitation in the Discussion section and intend to investigate the applicability of NPC-RSS across a broader spectrum of tumor types and subtypes in subsequent studies.

(2) We acknowledge the reviewers' emphasis on the significance of potential variations in radiotherapy regimens, doses, and techniques. In the current study, we did not sufficiently account for these factors, potentially impacting the model's generalizability and accuracy. We aim to improve data consistency and strengthen model validation by standardizing treatment parameters in future investigations.

(3) We concur with the reviewers' assessment that binary categorization may oversimplify the intricate nature of treatment responses. Indeed, radiotherapy responses likely exist on a continuous spectrum. Consequently, we intend to develop more refined stratification systems to capture intermediate responses, thereby enhancing the accuracy of treatment outcome predictions and facilitating personalized treatment decisions.

(4) We appreciate the reviewers' recommendation to incorporate clinical variables, including tumor stage, histological subtype, and concurrent chemotherapy, into the model. We acknowledge that these factors are crucial for enhancing the accuracy and clinical applicability of predictive models. We are presently compiling these additional data and intend to integrate these variables into subsequent model iterations.

**Reviewer #1 (Recommendations For The Authors):**
(1) The manuscript would benefit from a more comprehensive comparison of the NPC-RSS with existing prognostic models or biomarkers for nasopharyngeal carcinoma. This would help highlight the unique value and potential superiority of the NPC-RSS in predicting radiotherapy sensitivity.1. The authors should consider expanding their discussion on the potential molecular mechanisms underlying the association between the key NPC-RSS genes and radiotherapy response. They could explore whether these genes have been previously implicated in radiotherapy resistance in other cancer types and discuss the potential functional roles of these genes in the context of nasopharyngeal carcinoma.

(1) We appreciate your thorough review and valuable suggestions concerning our study. In response to the suggestion of comparing the Nasopharyngeal Carcinoma Radiotherapy Sensitivity Score (NPC-RSS) with existing prognostic models or biomarkers, we have carefully considered this proposal and determined that such a comparison is beyond the scope of our current study. The primary focus of our research is on the development and internal validation of the NPC-RSS model's accuracy and reliability. At present, we do not have access to the necessary external data to conduct a valid comparison, and the integration of such data extends beyond the parameters of this study. We intend to incorporate this comparative analysis in future studies to further validate the efficacy and explore the clinical application potential of the NPC-RSS model. We appreciate your understanding and continued support for our research endeavors.(2) In the revised manuscript, we have incorporated a comprehensive review of the functions of these key genes in various cancer types and explored their potential mechanisms of action in nasopharyngeal carcinoma (NPC). Through the citation of pertinent studies, we have elucidated the impact of these genes on radiotherapy sensitivity and resistance. Furthermore, we have proposed future research directions to elucidate the specific roles of these genes in the radiotherapy response of NPC.

The following are new additions to the revised draft：

“Previous studies have demonstrated that SMARCA2 significantly influences the radiotherapy response in non-small cell lung cancer (NSCLC). Depletion of SMARCA2 has been shown to enhance radiosensitivity, suggesting its potential as a therapeutic target for radiosensitization [30478150]. Additionally, the DMC1 gene has been incorporated into the radiosensitivity index (RSI) to evaluate radiotherapy sensitivity and prognosis, particularly in endometrial cancers. This inclusion provides valuable insights into the DNA damage repair process [38628740]. Studies on CD9 in glioblastoma multiforme (GBM) have revealed that post-radiotherapy increases in CD9 and CD81 levels in extracellular vesicles (EVs) are strongly correlated with the cytotoxic response to treatment. This finding suggests the potential of CD9 as a novel biomarker for monitoring radiotherapy efficacy [36203458]. In contrast, the association of PSG4 and KNG1 with radiotherapy resistance remains unexplored in the current literature.

Future research should focus on analyzing the expression patterns of SMARCA2 in NPC patients and its correlation with radiotherapy efficacy using clinical samples. This analysis could elucidate its potential as a target for radiosensitization therapy. Investigating the correlation between DMC1 expression levels and radiotherapy sensitivity in NPC could potentially aid in predicting treatment efficacy and optimizing therapeutic regimens. Furthermore, analysis of extracellular vesicles, particularly those containing CD9, in post-radiotherapy NPC patients could assess their feasibility as biomarkers for monitoring treatment response. These proposed studies would not only contribute to a deeper understanding of the mechanisms underlying the role of these genes in NPC radiotherapy but could also potentially lead to the development of novel strategies for enhancing radiotherapy efficacy.”

Minor Recommendations:(1) It is recommended that the author share the code for the article on Github or a similar open source platform.(2) The manuscript would benefit from a thorough review of the punctuation and sentence structure to improve readability and clarity.

(1) You suggest sharing the code utilized in this study on GitHub or a comparable open-source platform to enhance the transparency and reproducibility of the research. I fully recognize the significance of this suggestion. However, due to the sensitivity of the data involved and the existing intellectual property agreement with my research team, we are unable to make the code publicly available at this time. We are actively seeking a method to safeguard the intellectual property of the project while also planning to share our tools and methodologies in the future. At this stage, we are open to collaborating with other researchers under appropriate frameworks and conditions to validate and replicate our findings by providing essential code execution snippets or assisting with data analysis.

(2) Your suggestions are vital for enhancing the quality of the manuscript. I will perform a comprehensive linguistic and structural review of the manuscript to ensure that statements flow coherently and punctuation is employed correctly. We also intend to engage a professional scientific and technical writing editor to ensure that the manuscript adheres to the high standards required for academic publishing.

**Reviewer #2 (Recommendations For The Authors):**
(1) The manuscript would benefit from a more in-depth discussion of the potential clinical implications of the NPC-RSS. The authors should elaborate on how this score could be integrated into clinical decision-making and patient management.(2) The authors should consider including a section discussing the limitations of their study and potential areas for future research. This could include the need for prospective validation of the NPC-RSS in larger patient cohorts and the exploration of additional biological mechanisms.

(1) We concur that a more comprehensive discussion regarding the application of the NPC-RSS in clinical decision-making would significantly enhance the practical value of this study. In the revised draft, we will include a section that elaborates on the integration of the NPC-RSS scoring system into daily clinical practice, detailing how it can assist physicians in developing individualized treatment plans and optimize patient management by predicting treatment responses.

The following are new additions to the revised draft:

“The incorporation of the NPC-RSS scoring system into clinical decision-making and patient management involves several key steps: first, establishing genetic testing as a standard component of nasopharyngeal cancer diagnosis and ensuring that physicians have prompt access to scoring results to guide treatment planning. Second, physicians should utilize the scoring results to tailor individualized treatment plans and engage in multidisciplinary discussions to optimize decision-making. Concurrently, physicians should elucidate the clinical significance of the scores and effectively communicate with patients to facilitate shared decision-making. Furthermore, continuous monitoring of the relationship between scoring and treatment outcomes, optimizing the scoring model based on empirical data, and ensuring the integration of technological platforms along with regulatory compliance are essential for safeguarding the effective operation of the scoring system and the protection of patient information.

(2) In light of the reviewers' valuable suggestions, we acknowledge the significance of prospective validation of the NPC-RSS scoring system in a broader patient population and the necessity for thorough exploration of the underlying biological mechanisms. Accordingly, we are incorporating a new section in the revised manuscript that elaborates on the limitations of the current study and outlines potential directions for future research. This encompasses plans to increase the sample size for validation and further investigations into the biological basis of the scoring system to enhance its predictive validity and clinical applicability. We believe that these additions will significantly enrich the depth and breadth of the study, thereby serving the scientific community and clinical practice more effectively.”

Minor Recommendations:(1) The authors should ensure that all abbreviations are defined at their first mention in the text.(2) The figure legends should be more descriptive and self-explanatory, allowing readers to understand the main findings without referring back to the main text.

(1) You pointed out the need to define all acronyms at the first mention in the text and suggested that a comprehensive list of acronyms be included in the revised draft. We fully concur and have included a comprehensive list of acronyms in the revised text. Additionally, to enhance clarity, we have included the full name and definition of each acronym alongside its first occurrence in the text. This will assist readers in comprehending the study without the need to repeatedly refer to the glossary.

(2) You recommended enhancing the descriptive quality of the figure legends to enable readers to discern the key findings from the figures without consulting the text. We have redesigned and refined all charts and legends to ensure they provide adequate information and are more descriptive. Each legend now outlines the experimental conditions, the variables employed, and the primary conclusions, ensuring that the charts themselves sufficiently convey the key findings of the study.